# Multispectral and SAR Image Fusion Based on Laplacian Pyramid and Sparse Representation

**Hai Zhang** [1] , **Huanfeng Shen** [1,2,*], **Qiangqiang Yuan** [3] **and Xiaobin Guan** [1]

1   School of Resource and Environmental Sciences, Wuhan University, Wuhan 430079, China; haizhang@whu.edu.cn (H.Z.); guanxb@whu.edu.cn (X.G.)
2   Collaborative Innovation Center of Geospatial, Wuhan University, Wuhan 430079, China
3   School of Geodesy and Geomatics, Wuhan University, Wuhan 430079, China; qqyuan@sgg.whu.edu.cn
*   Correspondence: shenhf@whu.edu.cn; Tel.: +86-27-6877-8375

**Abstract:** Complementary information from multi-sensors can be combined to improve the availability and reliability of stand-alone data. Typically, multispectral (MS) images contain plentiful spectral information of the Earth's surface that is beneficial for identifying land cover types, while synthetic aperture radar (SAR) images can provide abundant information on the texture and structure of target objects. Therefore, this paper presents a fusion framework to integrate the information from MS and SAR images based on the Laplacian pyramid (LP) and sparse representation (SR) theory. LP is performed to decompose both the multispectral and SAR images into high-frequency components and low-frequency components, so that different processing strategies can be applied to multi-scale information. Low-frequency components are merged based on SR theory, whereas high-frequency components are combined based on a certain activity-level measurement, identifying salient features. Finally, LP reconstruction is performed to obtain the integrated image. We conduct experiments on several datasets to verify the effectiveness of the proposed method. Both visual interpretation and statistical analyses demonstrate that the proposed method strikes a satisfactory balance between spectral information preservation and the enhancement of spatial and textual characteristics. In addition, a further discussion regarding the adjustability property of the proposed method shows its flexibility for further application scenarios.

**Keywords:** Laplacian pyramid; sparse representation; multi-sensor image fusion; image quality assessment; synthetic aperture radar

## 1. Introduction

Multi-sensor image fusion aims to combine complementary information from different sources to obtain improved results with greater quality and reliability [1,2]. Fusion of multispectral (MS) and synthetic aperture radar (SAR) images, as one important branch of multi-sensor image fusion, has been attracting increasing attention. Multispectral image reflects abundant information of spectral signature of the ground objects and thus is useful for distinguishing different land cover types, whereas it heavily depends on solar illumination and weather conditions. On the other hand, SAR operates as an all-weather, all-time earth observation system, capable of penetrating clouds, rain, smoke, and fog. Moreover, SAR image usually has good contrast and provides sufficient textural and structural information of observed objects. Therefore, the integration of multispectral and SAR images can effectively utilize the complementary information and achieve the full potential of both datasets, which can improve the accuracy and efficiency of remote sensing image interpretation and information extraction, as well as further applications [3–6].

Due to the differences in imaging mechanism and spectral characteristics, multi-source heterogeneous image fusion (e.g., optical and SAR image fusion) has always been a sophisticated and challenging topic. Therefore, the literature that focuses on pixel-level multispectral and SAR image fusion algorithms is limited and usually borrows the techniques

utilized in the field of pansharpening. To our knowledge, most existing methods can be grouped into four categories: component substitution (CS)-based methods, multi-resolution analysis (MRA)-based methods, model-based methods, and hybrid methods [7]. In recent years, the deep learning-based method has gradually come into people's vision [8–10]. It is immature at the current stage due to the lack of massive training datasets and the adaptability to various sensors [11–13], which is beyond the scope of this paper. The earliest methods are basically based on component substitution, which is the substitution of a component obtained from a certain transformation of MS bands with the high spatial resolution (HR) SAR image. Representative CS-based methods include intensity-hue-saturation (IHS) [14,15], principal component analysis (PCA) [16,17], Gram–Schmidt (GS) [18], and Brovey [19]. CS-based methods can remarkably enhance the spatial features of MS images but usually cause more severe spectral distortion than in the pansharpening task. This is mainly because the assumption that the substituted component and high spatial resolution image have a strong correlation may be unreasonable in multispectral and SAR image fusion. The MRA-based methods rely on the injection of high-frequency information that is extracted from the high spatial resolution SAR image into the MS image via MRA tools. Popular MRA-based methods comprise wavelet transform (WT) [20], high-pass filter (HPF) [21], à trous wavelet transform (AWT) [22] and Laplacian pyramid (LP) [23]. Since only the spatial contents of MS images are changed, the spectral information is well preserved. Thus, MRA-based methods generally perform better than CS-based methods in the conservation of spectral properties. However, MRA-based methods unavoidably lose spatial characteristics contained in the low frequency of source images [24]. In addition, MRA-based approaches are sensitive to aliasing and misregistration, which is vulnerable to blocking artifacts and blur effects [25]. Model-based methods are also extended for the integration of MS and SAR images, which are originally proposed for the pansharpening tasks [26,27]. These methods may cause less registration errors but usually lead to a higher computational complexity.

According to the aforementioned introduction of the three groups of methods, they all have their own advantages and disadvantages. Naturally, hybrid methods that incorporate the merits of three groups of methods were proposed. Alparone et al. [28] combined the generalized IHS transform and à trous wavelet decomposition to inject panchromatic and SAR features into MS images. Chibani [29] conjointly used the modified Brovey transform (MBT) and à trous wavelet decomposition to integrate SAR features into MS images. Hong et al. [30] developed a fusion method based on IHS transform and wavelet decomposition to fuse moderate spatial resolution MS imagery and high spatial resolution SAR image. Yin [31] proposed a fusion algorithm based on support value transform (SVT) and sparse representation for the purpose of target recognition. Shao et al. [32] combined the IHS transform and gradient transfer fusion (GTF) algorithm to maintain the spatial details from images of both MS and SAR. Generally, hybrid methods have become more and more popular because of their comprehensive abilities of dealing with different problems [7].

Since most of the previous studies simply exploited the elementary methods developed many years ago, complementary information contained in MS and SAR images can not be utilized simultaneously. Therefore, it is necessary to explore more advanced tools such as sparse representation (SR) to excavate the underlying information. Sparse representation theory has been successfully applied in various fields of computer vision and image processing [33–36]. Sparse representation theory assumes that natural signals, such as images, can be represented or approximately represented by a linear combination of a small number of atoms, which are columns of an over-complete dictionary [37]. Li et al. [38] first brought the SR theory into multi-focus image fusion in 2009. Subsequently, ever-growing interests can be seen in SR-based image fusion [39,40], especially in the remote sensing image fusion [41] and multi-modal image fusion [42,43]. The reason why SR theory works effectively in image fusion tasks is that sparse coefficients can well describe the salient features and structure information contained in images, which is immensely beneficial for image fusion [38]. Nevertheless, stand-alone SR-based image fusion approaches have

some drawbacks [24]. Firstly, the sparse coding stage in SR-based image fusion usually consumes a considerable computational burden, and the running time will grow sharply as the size of source images increases. Secondly, the sliding window technique employed in sparse representation may cause smoothness and missing of details, especially when the overlapped regions between adjacent patches are large.

Therefore, in this paper, we proposes a fusion framework combining the merits of MRA-based and SR-based methods to integrate multispectral and SAR images. Specifically, Laplacian pyramid (LP) is chosen as the multi-scale transform tool in our study since images can be decomposed into uncorrelated high-frequency channels without losing details such as edges and textures [44]. Considering the first weakness, the running time can be sharply reduced, as the sparse coding operation is only implemented in small-sized low-pass bands. For the second demerit, most spatial information is separated by the LP and preserved in the high-frequency components via effective fusion rule. Furthermore, this paper also explores the flexibility and adjustability of the proposed fusion framework for potential applications.

The remainder of this paper is organized as follows. Section 2 describes the proposed multispectral and SAR image fusion framework in detail. The experimental results and analyses are provided in Section 3. Discussion about the adjustability of fusion results is included in Section 4. Finally, Section 5 concludes this paper.

## 2. Proposed Fusion Framework

The conventional Laplacian-pyramid-based image fusion methods generally employ the "averaging" rule to merge the low-frequency components, which usually leads to the loss of contrasts and details. On the other hand, SR-based image fusion approaches are limited by the high computation expenses and may be smoothed due to the sliding window technique. In order to overcome the disadvantages of stand-alone methods, we combine the Laplacian pyramid and sparse presentation theory to integrate multispectral and SAR images via the proposed fusion method.

In this paper, we adopt a band-wise fusion strategy for each channel of multispectral images, that is to say each channel follows the same fusion procedures to be integrated with SAR image. (1) LP generation. First, multispectral and SAR image LPs are constructed. Both the multispectral and SAR images are decomposed into high-frequency components and low-frequency components. (2) High-frequency components fusion. For high-frequency components, an appropriate fusion rule is employed to maximally reserve the high-frequency features, such as edges, point targets, and lines. (3) Low-frequency components fusion. Subsequently, sparse representation theory is applied to low-frequency components fusion. (4) Reconstruction. Finally, a reconstruction process is implemented using fused high-frequency and low-frequency components according to the inverse Laplacian pyramid transform. The proposed fusion method is outlined in Figure 1.

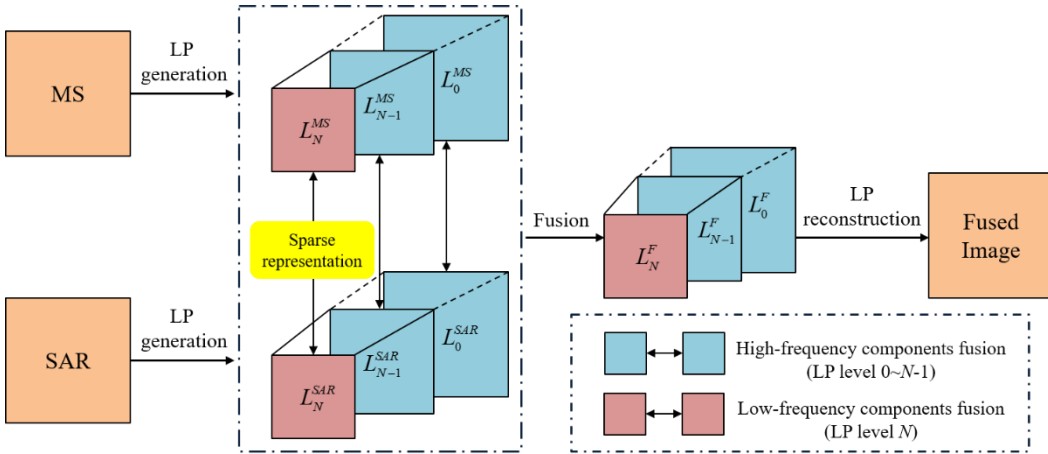

**Figure 1.** Schematic diagram of proposed fusion method.

### 2.1. LP Generation and Reconstruction

The Laplacian pyramid can be seen as a hierarchical image structure that represents an image as a sequence of bandpass sub-images at different spatial scales. This versatile data structure makes LP useful for image fusion [45]. Some multi-scale transform tools, such as wavelet transform, are critically subsampled and thus unavoidably create artifacts due to misregistration. Compared with them, LP is oversampled, without detail decimation, and therefore it is considerably appropriate for our study.

After the original multispectral image $I^{MS}$ and despeckled SAR image $I^{SAR}$ are aligned geometrically, LP decomposition is conducted to both MS and SAR images, and then the corresponding LPs are generated. The implementation details can refer to [46]. The N-level LPs of MS and SAR images are denoted as $\{L_l^{MS}\}_{l=0}^{N}$ and $\{L_l^{SAR}\}_{l=0}^{N}$, respectively. High-frequency information including textures and edges are maintained in a series of detail images $\{L_l^{MS}\}_{l=0}^{N-1}$ and $\{L_l^{SAR}\}_{l=0}^{N-1}$ at different spatial scales, while low-frequency information is stored in the approximation images $L_N^{MS}$ and $L_N^{SAR}$ at a much smaller scale compared to the original images.

Once the fused high-frequency $\{L_l^F\}_{l=0}^{N-1}$ and low-frequency $L_N^F$ components are obtained, the reconstruction process is performed following inverse Laplacian pyramid transform, and then the fused image $L_N^F$ is obtained. The reconstruction process can be expressed as:

$$I^F = L_0^F + \text{EXPAND}(L_1^F + \text{EXPAND}(L_2^F + \cdots + \text{EXPAND}(L_N^F))) \tag{1}$$

where the function EXPAND consists of up-sampling and interpolating operations [46].

### 2.2. High-Frequency COMPONENTS Fusion

High-frequency components are also known as detail images in LP, which contain abundant details such as textures and edges. The absolute value of each pixel in the detail image is set as activity level measurement, which represents the sharpness or edges of an image. Accordingly, a popular "max-absolute" fusion rule [47] is exploited for merging the detail images $\{L_l^{MS}\}_{l=0}^{N-1}$ and $\{L_l^{SAR}\}_{l=0}^{N-1}$. The fused high-frequency components $\{L_l^F\}_{l=0}^{N-1}$ are obtained according to Equation (2):

$$L_l^F(i,j) = \begin{cases} L_l^{MS}(i,j), & \text{if } |L_l^{MS}(i,j)| \geq |L_l^{SAR}(i,j)| \\ L_l^{SAR}(i,j), & \text{otherwise} \end{cases} \tag{2}$$

where $l = 0, 1, \cdots, N-1$, $0 \leq i < R_l$, $0 \leq j < C_l$, $R_l$, and $C_l$ are the row and column number of $L_l^{MS}$, $L_l^{SAR}$, and $L_l^F$.

### 2.3. Low-Frequency Component Fusion

The low-frequency components in LP are also called the approximation images, which approximately represent the source image in a smaller scale. The traditional multi-resolution analysis (MRA) fusion methods often adopt a simple average rule when the low-frequency information is fused. Since most of the energy in an image is concentrated in the low-frequency information, the averaging operation inevitably reduces the contrast of the fusion results [24]. Consequently, in the proposed low-frequency component fusion, sparse representation theory is utilized to maintain the information that may be lost in the LP-based fusion framework. The SR-based fusion method can extract the local structure information contained in the low-frequency components thanks to the powerful image feature expression ability of the over-complete dictionary [38]. Therefore, the energy of original images is maintained as much as possible, and the contrast of the fused image is improved. The flowchart of the low-frequency component fusion, based on sparse representation, is illustrated in Figure 2.

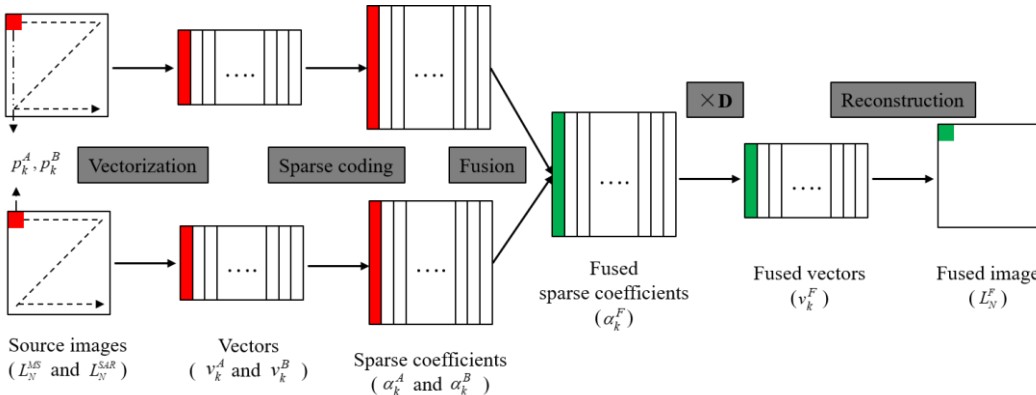

**Figure 2.** The flowchart of SR-based low-frequency component fusion.

For convenience, the low-frequency component $L_N^{MS}$ of MS image Laplacian pyramid is denoted as $A$, the low-frequency component $L_N^{SAR}$ of SAR image Laplacian pyramid is denoted as $B$, and the fused low-frequency component $L_N^F$ of fusion image Laplacian pyramid is denoted as $F$. The size of image $A$, $B$, and $F$ is $M \times N$. The SR-based low-frequency components fusion in this paper takes the following steps:

(1) Patch generation. To make full use of the local information of source images, the sliding window technique is applied to divide the source images $A$ and $B$ into image patches $p_k^A, p_k^B$ ($k = 1, 2, \cdots, K$ and $K = (M - \sqrt{n} + 1) \times (N - \sqrt{n} + 1)$) of size $\sqrt{n} \times \sqrt{n}$, starting from the top-left to the bottom-right with a fixed step length $s$.

(2) Vectorization. Then, image patches $p_k^A, p_k^B$ are rearranged to $n \times 1$ vectors $v_k^A, v_k^B$ in a column-wise way. Each vector is normalized to zero-mean via subtracting the mean value according to the following Equation (3), and the mean values are stored for subsequent reconstruction process [47],

$$\begin{cases} \hat{v}_k^A = v_k^A - \overline{v}_k^A \cdot 1 \\ \hat{v}_k^B = v_k^B - \overline{v}_k^B \cdot 1 \end{cases} \tag{3}$$

where $\overline{v}_k^A$ and $\overline{v}_k^B$ are the mean values of vectors $v_k^A$ and $v_k^B$, respectively, and 1 is a $n \times 1$ vector of all ones.

(3) Sparse coding. Calculate the sparse coefficients of vectors $\hat{v}_k^A$ and $\hat{v}_k^B$ according to Equation (4) using the simultaneous orthogonal matching pursuit (SOMP) algo-

rithm [48]. The SOMP algorithm is employed here for its high computing efficiency and suitability for image fusion,

$$
\begin{cases}
\boldsymbol{\alpha}_k^A = \underset{\alpha}{\arg\min} \|\boldsymbol{\alpha}\|_0 \ s.t. \|\hat{v}_k^A - \mathbf{D}\boldsymbol{\alpha}\|_2^2 \leq \varepsilon \\
\boldsymbol{\alpha}_k^B = \underset{\alpha}{\arg\min} \|\boldsymbol{\alpha}\|_0 \ s.t. \|\hat{v}_k^B - \mathbf{D}\boldsymbol{\alpha}\|_2^2 \leq \varepsilon
\end{cases}
\tag{4}
$$

where $\|\boldsymbol{\alpha}\|_0$ denotes the number of nonzero elements in $\boldsymbol{\alpha}$, $\mathbf{D}$ refers to a pre-defined dictionary, and $\varepsilon$ is the error tolerance.

(4) Coefficient fusion. The activity level measurement and fusion rule are two important issues in image fusion tasks [47]. In this paper, the absolute value of the sparse coefficient is chosen to describe the activity level, and the popular max-absolute rule is selected as the fusion rule to combine the corresponding sparse coefficients. The detailed fusion process can be described by Equation (5):

$$
\boldsymbol{\alpha}_k^F(t) = \begin{cases}
\boldsymbol{\alpha}_k^A(t), \ \text{if } |\boldsymbol{\alpha}_k^A(t)| \geq |\boldsymbol{\alpha}_k^B(t)| \\
\boldsymbol{\alpha}_k^B(t), \ \text{otherwise}
\end{cases}
\tag{5}
$$

where $\boldsymbol{\alpha}_k^A(t)$ and $\boldsymbol{\alpha}_k^B(t)$ are the $t$-th element in $\boldsymbol{\alpha}_k^A$ and $\boldsymbol{\alpha}_k^B$, respectively.

(5) Vector reconstruction. The fused sparse vector $\hat{v}_k^F$ is obtained via the fused sparse coefficient $\boldsymbol{\alpha}_k^F$ multiplied by the same dictionary used in Step (3). The local mean subtracted in Step (2) is added back, and the final fused vector $v_k^F$ is obtained.

$$
\hat{v}_k^F = \mathbf{D}\boldsymbol{\alpha}_k^F
\tag{6}
$$

$$
\overline{v}_k^F = \frac{1}{2}(\overline{v}_k^A + \overline{v}_k^B)
\tag{7}
$$

$$
v_k^F = \hat{v}_k^F + \overline{v}_k^F \cdot 1
\tag{8}
$$

(6) Final reconstruction. Every fused sparse vector $v_k^F$ is reshaped to a $\sqrt{n} \times \sqrt{n}$ patch and placed in the corresponding position in the fused image *F*. As the patches may be overlapped, the same pixel in the source image may appear in multiple patches. In other words, one position in *F* may relate to multiple patches. Therefore, each pixel's value in the fused image *F* is the average value of the corresponding elements in all related patches. Finally, the fused low-frequency component $L_N^F$ is obtained.

## 3. Experiments

### 3.1. Experiment Settings

3.1.1. Data Description

In order to verify the validity and effectiveness of the proposed fusion framework, we selected three pairs of airborne multispectral images and X-band intensity images of airborne SAR AeS-1 [49,50] for experiments, covering the Trudering region of Munich in Germany. The original multispectral and SAR images have the same spatial resolution of 1.5 m and are co-registered. In this paper, multispectral images were spatially degraded to 6 m in order to verify the spatial enhancement of various algorithms. An advanced SAR despeckling algorithm [51] was selected to reduce the impact of speckle noise. Subsequently, multispectral images were resampled to the same size of SAR images.

As shown in Figure 3, area 1 is featured with agricultural fields and field ridges, area 2 is dominated by buildings and grasslands, and area 3 contains more diversified land cover types. The selected areas feature abundant land cover types and ground features, with different shapes, sizes, and orientations that are appropriate for evaluation of the fusion methods in terms of spectral preservation and spatial enhancement.

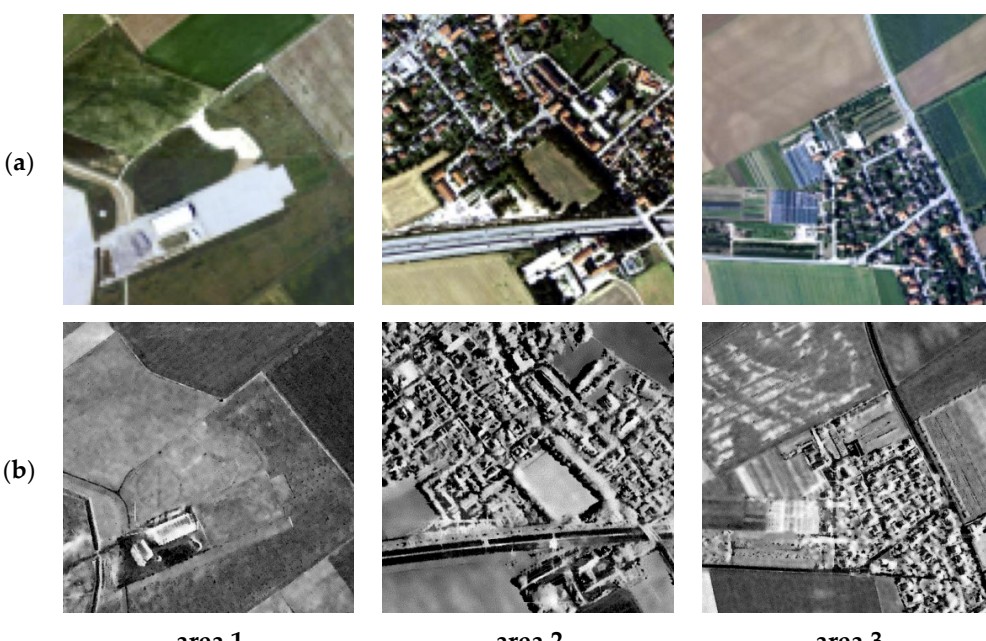

area 1         area 2         area 3

**Figure 3.** Multispectral and SAR images used for fusion experiments, located in the Trudering region of Munich in Germany. (**a**) Airborne multispectral images; (**b**) airborne SAR intensity images.

### 3.1.2. Evaluation Metrics

Both visual inspection and quantitative analysis are included to evaluate the performance of all the fusion methods. Several popular evaluation indicators are employed to quantitatively analyze the fusion results in terms of spectral preservation, spatial enhancement, and the amount of information. Correlation coefficient (CC) and spectral angle mapper (SAM) [52] are selected to evaluate the spectral fidelity between the spatially degraded HR multispectral and fused image. Spatial correlation coefficient (SCC) [53], structural similarity (SSIM) [54], and average gradient (AG) [50] are chosen as indicators measuring the spatial enhancement of various fusion approaches. Feature mutual information (FMI) [55] is used to assess the amount of information transferred from the source images. Detailed descriptions can be found in the literatures listed above. All the metrics are calculated as average values across multi bands, except that SAM is averaged over all the pixels.

In order to synthetically compare the statistical results, we also define a comprehensive index (CI) considering spectral preservation, spatial enhancement, and information incorporation simultaneously, which is formulated as:

$$\mathrm{CI} = \frac{\sum\limits_{i} \omega_i X_i}{\sum\limits_{i} \omega_i} \tag{9}$$

where $X_i$ denotes the $i$-th evaluation metric, and $\omega_i$ represents the corresponding weight of the evaluation metric.

### 3.1.3. Comparison Methods

Five typical fusion methods covering CS-based, MRA-based, and hybrid methods were chosen for comparison with the proposed Laplacian pyramid and sparse representation-based (LPSR) method, including IHS fusion method [14], BTH fusion method [56], Laplacian pyramid (LP)-based fusion method [23], AWLPR fusion method [57], and IHS+Wavelet (IW) fusion method [30]. In addition, we also give the fusion results employing a competitive pansharpening method (GFTD) proposed by [58] with the best parameters authors suggested. The IW fusion method used the 'Db4' filter. The patch size in the sparse coding

stage of the proposed method was $8 \times 8$, and the sliding window's step size was set to 1. The DCT dictionary was chosen for generality in this work, and the error tolerance $\varepsilon$ in SR-based low-frequency components fusion was set to 0.01. The source code of the proposed method can be downloaded at http://sendimage.whu.edu.cn/send-resource-download/ (accessed on 27 January 2022).

### 3.2. Experimental Results

Figure 4 shows the fusion results of different methods in area 1, which features agricultural lands. The integration of MS and SAR images can effectively enhance the transitions and boundaries between buildings, roads, and cultivated lands and retain the colors at the same time. Figure 4a,b shows the HR SAR image and up-sampled LR multispectral image, which are the input images for various fusion methods. The HR multispectral image is provided for spectral evaluation (but not the ground truth), as shown in Figure 4j. Compared with Figure 4b, Figure 4c–h demonstrates that all the fusion methods improve the sharpness of MS image and effectively enhance the spatial and texture features. As can be seen from Figure 4c,d, the spatial enhancement of IHS and BTH fusion is obvious, but the colors also appear unnatural. The spectral distortions of agricultural land in the upper right corner and the building in the lower left are serious. At the same time, serious noises can be seen in the red circle marked in the upper right corner in Figure 4c,d. Compared with the original HR multispectral image, the fusion results of AWLPR and IW fusion (see Figure 4f,g) perform well in spectral preservation but integrate limited spatial and texture features. Besides, obvious artifacts can be seen near the boundaries of ground features. Figure 4h (GFTD fusion method) exhibits appealing visual performance, while it suffers from aberrant colors in the areas marked with yellow arrows. Figure 4e,i shows that the LP-based and proposed methods strike the balance between spectral preservation and spatial enhancement. The colors of these two methods are comparable to the HR multispectral image, and the buildings and boundaries are markedly enhanced. Moreover, the noises and artifacts are effectively avoided. Compared with the LP-based method, the fusion result of the LPSR method looks sharper in the region marked with a green rectangle, and overall, the image is clearer, with stronger contrast.

The fact that the ground truth image of multispectral and SAR image fusion does not exist makes the quantitative evaluation challenging and less reliable. Therefore, we just give the qualitative comparison for supplements. In order to compare the spectral fidelity of different fusion results, the first property of Wald's protocol [59] is checked via calculating the spectral indexes between the spatially degraded HR multispectral image and the degraded fusion images (reduced resolution). For testing the second and third property of Wald's protocol, we compared the differences of original HR multispectral and fused images (full resolution) in the spectral aspect. Compared with the pansharpening field, estimation of the similarity between the fused and HR SAR image in spatial details is much more frequently used and important. In order to analyze the spatial quality, Zhou's procedure [53] is adopted to measure the spatial similarity between SAR and the fused images. Additionally, a comprehensive metric based on information theory [55] is used to calculate the amount of gradient information conducted from both the input multispectral and SAR images to the fused images.

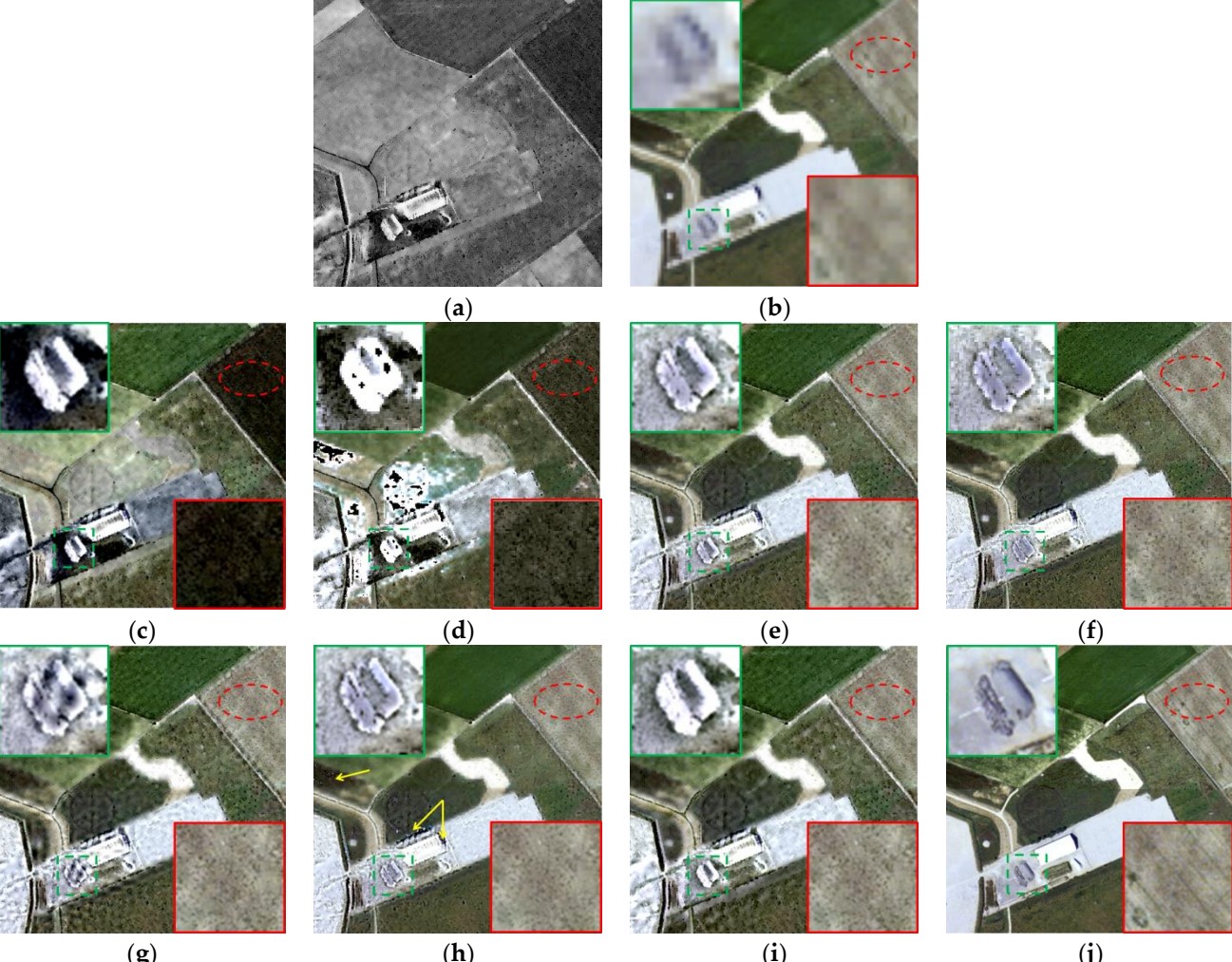

**Figure 4.** Fusion results of different methods in area 1. (**a**) HR SAR image; (**b**) up-sampled LR multispectral image; (**c**) IHS fusion; (**d**) BTH fusion; (**e**) LP-based fusion; (**f**) AWLPR fusion; (**g**) IW fusion; (**h**) GFTD fusion; (**i**) proposed LPSR fusion; (**j**) original HR multispectral image.

Table 1 displays the quantitative evaluation results of area 1. Overall, the statistical indicators are in agreement with the visual inspection. In terms of two spectral fidelity evaluation indicators, CC and SAM, the competitive pansharpening fusion methods GFTD and AWLPR have the strongest spectral preservation abilities, no matter if in the reduced or full spatial resolution. LP-based, IW, and the proposed LPSR fusion methods are a little bit weaker than the GFTD and AWLPR methods. IHS and BTH fusion methods have the worst results in all spectral fidelity indexes. Moreover, the CC of IHS method is almost zero, which indicates that the fusion result has no correlation with the multispectral image because of losing too many spectral characteristics. With regard to the spatial enhancement, IHS fusion method performs the best in SCC and SSIM indexes, which demonstrates that IHS fusion method integrates more spatial and texture features so that the fused images are closer to the HR SAR image. The SCC and SSIM of the proposed method are lower than the IHS method but better than the other methods. The BTH and GFTD method are the worst in terms of SCC and SSIM indicators because the intensity and histogram-matched SAR image have low contrast and limited details due to the low correlation between MS and SAR images. This phenomenon is extraordinarily common when simply borrowing the method from the pansharpening field. The LP-based, AWLPR, and IW methods lose edges because some structures and contents contained in the low frequency of SAR are discarded. The AGs of all the methods are higher than that of the original LR multispectral

image (5.0008), which proves that the multispectral and SAR image fusion can enhance the spatial and texture features of the original MS image. FMI indicates that IHS and our method integrate more gradient information into the fused image from the source images.

**Table 1.** Quantitative evaluation results of different fusion methods in area 1.

| Indexes | IHS | BTH | LP | AWLPR | IW | GFTD | LPSR | Ideal |
|---|---|---|---|---|---|---|---|---|
| CC [1] | 0.0418 | 0.4210 | 0.9620 | 0.9753 | 0.9608 | 0.9751 | 0.9235 | 1 |
| | 0.0247 | 0.3566 | 0.8663 | 0.8917 | 0.8598 | 0.8964 | 0.8238 | 1 |
| SAM [2] | 4.7668 | 2.9880 | 1.3685 | 1.0067 | 1.2048 | 1.0249 | 1.9790 | 0 |
| | 7.5331 | 5.6051 | 5.1160 | 4.3680 | 4.7799 | 4.3835 | 5.5635 | 0 |
| SCC | 0.9746 | 0.6436 | 0.9504 | 0.9506 | 0.9531 | 0.8714 | 0.9560 | 1 |
| SSIM | 0.8998 | 0.6862 | 0.7023 | 0.6632 | 0.6656 | 0.6572 | 0.7340 | 1 |
| AG | 20.452 | 19.7403 | 21.0458 | 20.6936 | 21.0910 | 16.1392 | 20.8458 | $+\infty$ |
| FMI | 0.5161 | 0.4061 | 0.4794 | 0.4678 | 0.4803 | 0.4652 | 0.5096 | 1 |

[1,2] The first row of indicator CC/SAM shows the spectral fidelity in the reduced spatial resolution, in order to check the first property of Wald's protocol. The second row corresponds to the results in the full spatial resolution, which verifies the second and third property of Wald's protocol.

Figure 5 exhibits the fusion results in area 2. Area 2 is dominated by buildings and roads. Consequently, the goal of image fusion is to improve the spatial resolution and effectively enhance the interpretability of the image. As can be seen from Figure 5, all the methods enhance the contrast of MS image and make the image more interpretable, which verifies the potentials of MS and SAR image fusion in urban areas. Figure 5c,d indicates that the IHS and BTH fusion methods can significantly improve the spatial resolution of the images. However, the fused images of IHS and BTH methods lose too many spectral characteristics of the multispectral image, which affects the visual interpretation. Figure 5e–h suggests that LP-based, AWLPR fusion, IW fusion, and GFTD methods can better maintain the spectral contents of MS image, whereas the fused images are blurred. In addition, the results of IW methods introduce visible artifacts in the boundaries. Again, GFTD method suffers from weird spectral distortion. In contrast, Figure 5i is the best result intuitively, which prevents the considerable disturbance of spectral contents and severe artifacts. Compared with Figure 5e, the proposed LPSR method shows better maintenance of spatial details and makes the image more interpretable than the LP-based method. Experimental results indicate that the proposed method achieves a satisfactory trade-off between spectral maintenance and spatial enhancement, which is the most suitable one in urban areas.

Table 2 displays the quantitative evaluation results of area 2, from which similar conclusions to area 1 can be drawn. AWLPR and GFTD methods perform the best in terms of indicators measuring spectral quality, followed by LP-based, IW, and the proposed LPSR methods. Again, the IHS method performed the worst in spectral preservation because it introduce excessive SAR features, including speckle noise. As to the indexes measuring spatial enhancement, IHS is the highest in SCC and SSIM. The proposed method is just inferior to the IHS method in SCC and SSIM and notably outperforms LP-based, AWLPR, IW, and GFTD methods. The SCC and SSIM indicators demonstrate that the proposed LPSR fusion method also has a satisfactory spatial enhancing ability. The AGs of all the methods are higher than that of the original LR multispectral image (13.6610), which proves the advantage of doing such fusion. Similarly, IHS has the highest FMI values, followed by our method, which are superior to the other fusion methods.

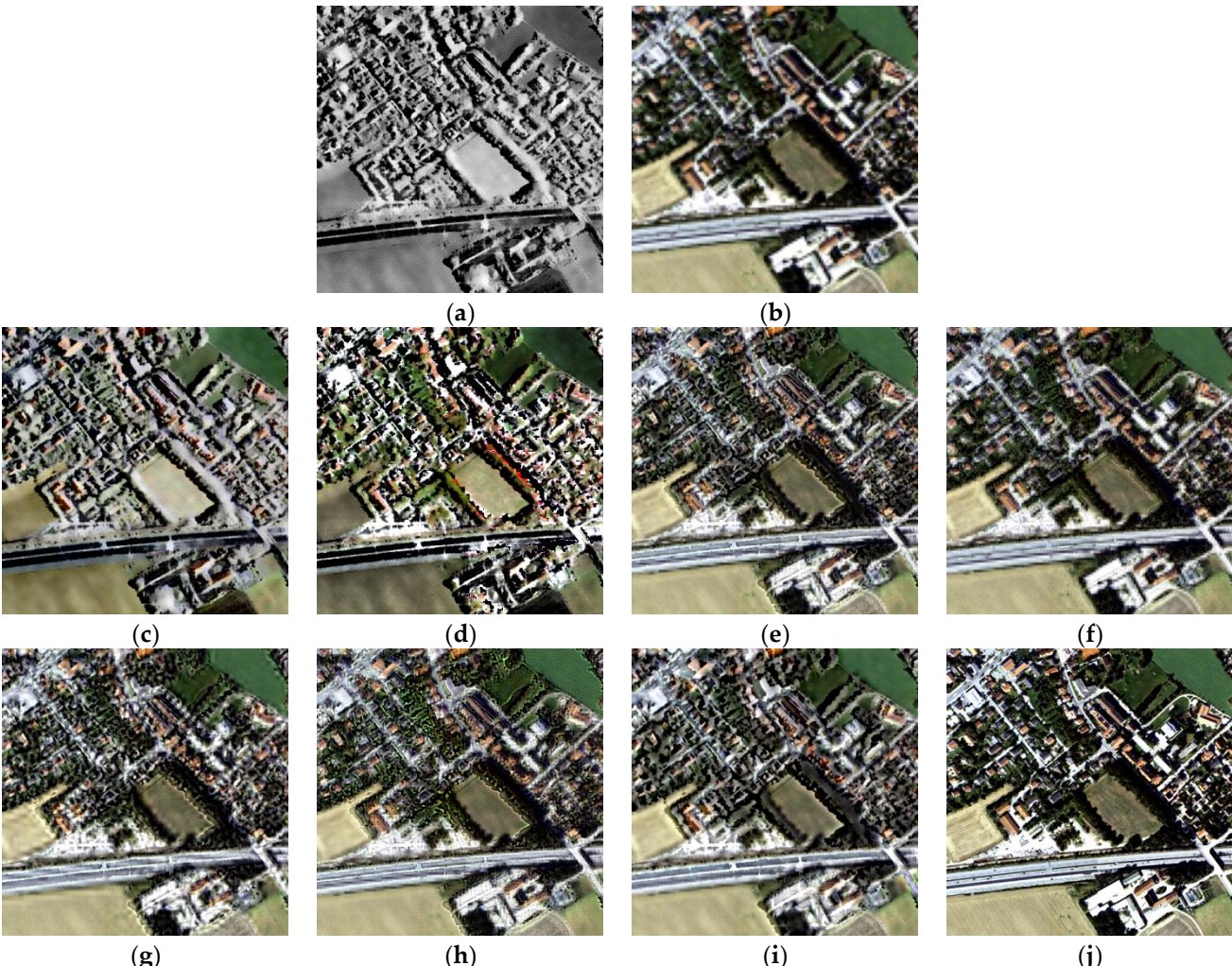

**Figure 5.** Fusion results of different methods in area 2. (**a**) HR SAR image; (**b**) up-sampled LR multispectral image; (**c**) IHS fusion; (**d**) BTH fusion; (**e**) LP-based fusion; (**f**) AWLPR fusion; (**g**) IW fusion; (**h**) GFTD fusion; (**i**) proposed LPSR fusion; (**j**) original HR multispectral image.

**Table 2.** Quantitative evaluation results of different fusion methods in area 2.

| Indexes | IHS | BTH | LP | AWLPR | IW | GFTD | LPSR | Ideal |
|---|---|---|---|---|---|---|---|---|
| CC [1] | −0.1951 | 0.0304 | 0.8679 | 0.9288 | 0.8582 | 0.9044 | 0.7401 | 1 |
| | −0.1832 | −0.0142 | 0.6846 | 0.8029 | 0.6598 | 0.7267 | 0.5603 | 1 |
| SAM [2] | 6.4627 | 3.3283 | 3.8941 | 2.6314 | 3.7114 | 2.7748 | 4.6309 | 0 |
| | 14.2631 | 9.6311 | 10.4075 | 9.0525 | 10.2610 | 9.4752 | 10.8905 | 0 |
| SCC | 0.9842 | 0.5734 | 0.8498 | 0.5453 | 0.8821 | 0.8571 | 0.9150 | 1 |
| SSIM | 0.9681 | 0.7428 | 0.4046 | 0.1429 | 0.3466 | 0.3242 | 0.5777 | 1 |
| AG | 24.6415 | 33.3441 | 25.7552 | 19.4118 | 24.7351 | 26.2544 | 23.4259 | +∞ |
| FMI | 0.5130 | 0.3785 | 0.4033 | 0.3974 | 0.4071 | 0.4217 | 0.4525 | 1 |

[1,2] The first row of indicator CC/SAM shows the spectral fidelity in the reduced spatial resolution, in order to check the first property of Wald's protocol. The second row corresponds to the results in the full spatial resolution, which verifies the second and third property of Wald's protocol.

Figure 6 displays the fusion results in area 3. As shown in Figure 6, all the methods improve the spatial resolution of the LR multispectral image to some extent, indicating that these methods more or less integrate the spatial details of SAR image into the fused images.

Figure 6c,d shows that the results of IHS and BTH fusion are similar, which is to say the results look sharper but suffer from severe color changes. Compared to the original HR multispectral image, the roads become black and the color of vegetation changes greatly. IHS and BTH fusion methods put too strong of an emphasis on spatial detail enhancement, resulting in the lack of necessary interpretability of the fusion results. Figure 6e–g shows that LP-based, AWLPR, and IW fusion methods maintain the spectral characteristics better. For example, the colors of grasslands and roads appear more natural and are closer to the colors in HR multispectral image. This is because LP-based, AWLPR, and IW fusion methods only incorporate the high-frequency information of SAR image and leave the low-frequency components of MS image unchanged, resulting in small spectral distortion. Compared with Figure 6e, obvious artifacts can be seen near the linear objects such as roads in Figure 6g. This phenomenon is common in WT-based fusion methods. By contrast, LP-based fusion methods abandon the decimation operator in the spatial frequency domain without losing the spatial connectivity of the edges and textures, and thus effectively preserve the spatial information of the original images [44]. AWLPR and GFTD method (Figure 6f,h) obtains the spectrally closest result compared to the HR multispectral image, while the fused image looks blurry and with low contrast. As shown in Figure 6i, the proposed LPSR method not only maintains a satisfactory spectral fidelity, but also integrates abundant spatial and texture features.

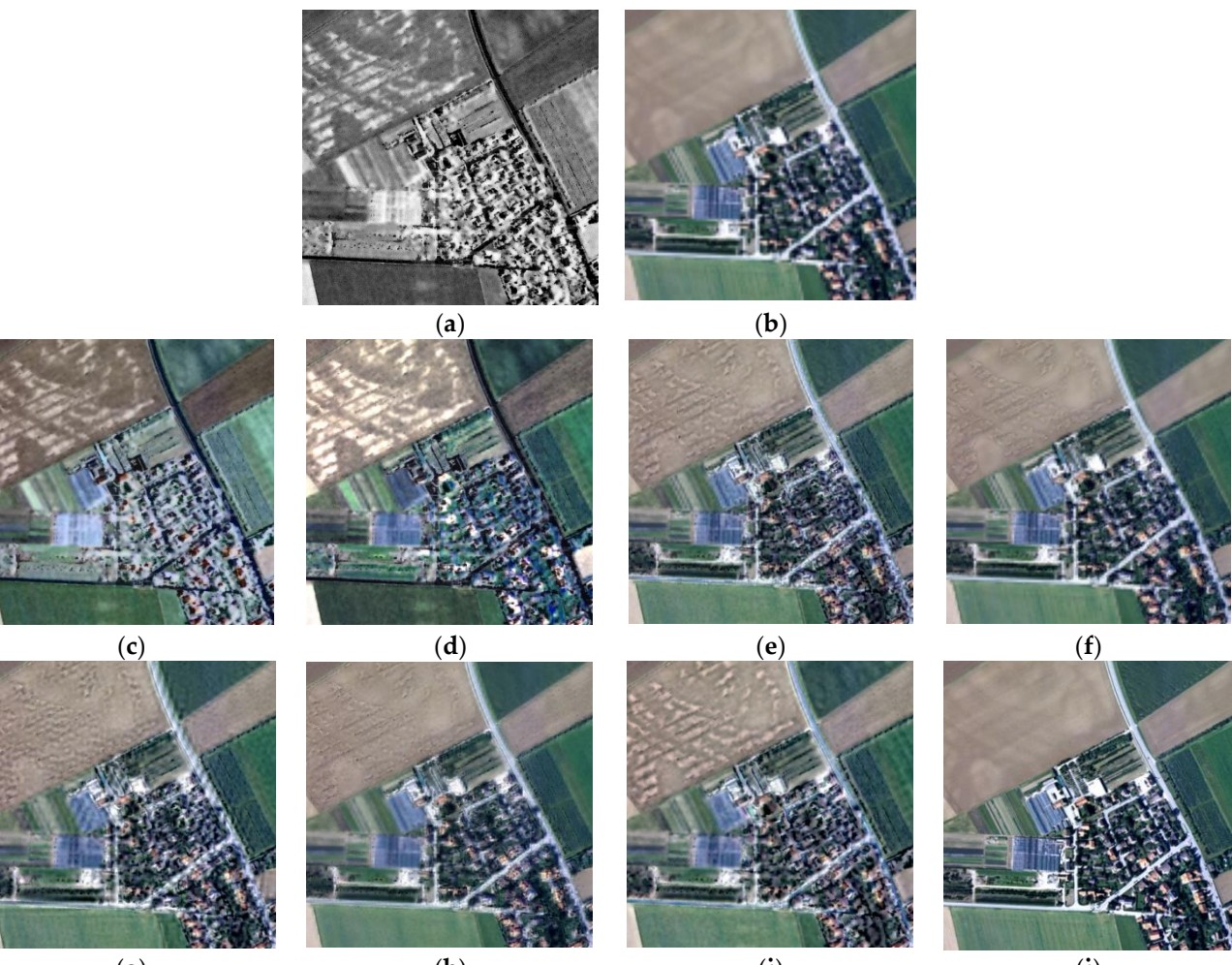

**Figure 6.** Fusion results of different methods in area 3. (**a**) HR SAR image; (**b**) up-sampled LR multispectral image; (**c**) IHS fusion; (**d**) BTH fusion; (**e**) LP-based fusion; (**f**) AWLPR fusion; (**g**) IW fusion; (**h**) GFTD fusion; (**i**) proposed LPSR fusion; (**j**) original HR multispectral image.

Table 3 shows the quantitative evaluation results of area 3. IHS method performs the worst in terms of the two spectral evaluation indexes CC and SAM. The other methods perform much better in spectral assessment compared to his. Among them, AWLPR and GFTD achieve the best results in spectral fidelity. Indicators SCC and SSIM show that the proposed method has a better spatial enhancement ability than BTH, LP-based, AWLPR, IW, and GFTD methods and is only worse than the IHS method. The AGs of all the methods are higher compared to that of the original LR multispectral image (7.7424), indicating that the fused images are clearer and contain more abundant spatial and texture information. Similar to the aforementioned two areas, the FMI values of IHS and the proposed method are much higher than other methods.

**Table 3.** Quantitative evaluation results of different fusion methods in area 3.

| Indexes | IHS | BTH | LP | AWLPR | IW | GFTD | LPSR | Ideal |
|---------|-----|-----|-----|-------|-----|------|------|-------|
| CC [1] | −0.1942 | 0.2846 | 0.8796 | 0.9335 | 0.8411 | 0.9212 | 0.7690 | 1 |
|  | −0.1923 | 0.2008 | 0.6712 | 0.7926 | 0.6151 | 0.7481 | 0.5668 | 1 |
| SAM [2] | 4.1220 | 2.1920 | 2.3637 | 1.5717 | 2.0341 | 1.6525 | 3.0498 | 0 |
|  | 7.5990 | 5.6608 | 6.3445 | 4.9756 | 5.8844 | 5.1918 | 6.9901 | 0 |
| SCC | 0.9972 | 0.9199 | 0.9138 | 0.6163 | 0.9458 | 0.9139 | 0.9629 | 1 |
| SSIM | 0.9583 | 0.7920 | 0.4885 | 0.2115 | 0.4376 | 0.3468 | 0.6534 | 1 |
| AG | 16.9149 | 17.2255 | 17.8394 | 11.6247 | 17.1778 | 13.91111 | 16.9217 | $+\infty$ |
| FMI | 0.5164 | 0.4536 | 0.4347 | 0.4223 | 0.4326 | 0.4369 | 0.4873 | 1 |

[1,2] The first row of indicator CC/SAM shows the spectral fidelity in the reduced spatial resolution, in order to check the first property of Wald's protocol. The second row corresponds to the results in the full spatial resolution, which verifies the second and third property of Wald's protocol.

The evaluation metrics for CI are correlation coefficient on the full spatial resolution ($CC_{full}$), structural similarity (SSIM), and feature mutual information (FMI). We selected them because they measure the spectral preservation, spatial enhancement, and information incorporation abilities. Moreover, the value ranges of them are similar. The weights are equal for them because we considered the abilities of spectral preservation, spatial enhancement and information incorporation equally in this paper. Figure 7 illustrates the CI values of all the methods in three areas, from which we can get a more intuitive impression on the quantitative performance. The proposed LPSR fusion method obtains the highest CI values in all the three areas, which indicates that our method achieves the best performances from a comprehensive angle considering spectral, spatial, and information characteristics. In addition, the order of CI values highly accords with visual inspection results, which validates the effectiveness of the proposed index.

To verify the stability of the results disregarding the variability of input images, we conducted experiments on a larger set of images, which consists of 116 pairs of multispectral and SAR images of size 100 × 100. Table 4 exhibits the average performance and standard deviation of each method across all the images. CS-based methods (IHS and BTH) behave the worst in terms of spectral information preservation (indexes CC and SAM), while MRA-based method AWLPR achieves the best results in spectral fidelity indexes CC and SAM. Regarding the spatial enhancement performance, CS-based method IHS introduces the most spatial and texture features in terms of SCC and SSIM, while MRA-based methods (LP and AWLPR) behave poorly in maintaining spatial characteristics. All methods improve the average gradients of the LR multispectral images (9.4163), which demonstrates that the MS-SAR fusion can powerfully incorporates SAR features into MS images and improve the information richness of remotely sensed images. The extraordinarily high value of AG from BTH may owe to the outliers in heterogeneous areas (such as in Figure 4d). The proposed method strikes a satisfactory trade-off between spectral and spatial preservation, which can be validated by the CI value in Figure 8.

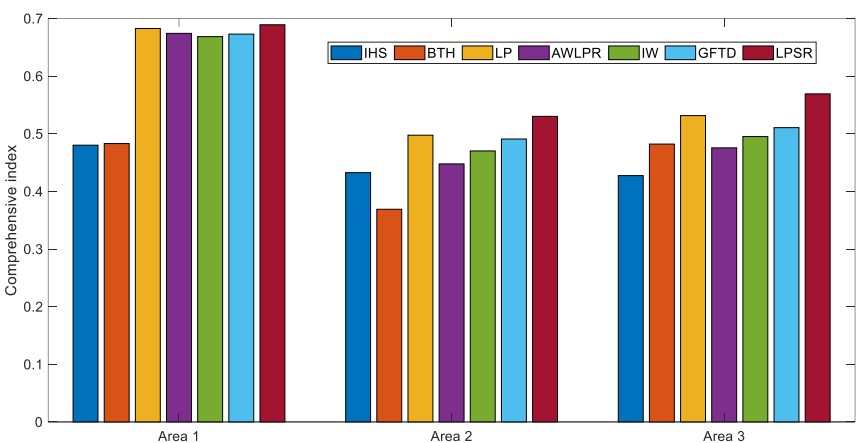

**Figure 7.** Bar chart of the comprehensive index.

**Table 4.** Quantitative analysis on a large set of images.

| Methods | CC [1] | SAM [2] | SCC | SSIM | AG | FMI |
|---|---|---|---|---|---|---|
| IHS | $-0.1263 \pm 0.2300$ | $7.7138 \pm 5.9478$ | $0.9551 \pm 0.0563$ | $0.8416 \pm 0.1150$ | $18.5431 \pm 8.5031$ | $0.5029 \pm 0.0314$ |
| BTH | $0.1577 \pm 0.2531$ | $6.8539 \pm 4.2926$ | $0.6217 \pm 0.2489$ | $0.5792 \pm 0.1902$ | $28.9911 \pm 12.5180$ | $0.3723 \pm 0.0569$ |
| LP | $0.6617 \pm 0.1131$ | $6.1318 \pm 4.1853$ | $0.8661 \pm 0.0597$ | $0.5488 \pm 0.1319$ | $20.1218 \pm 8.7419$ | $0.4266 \pm 0.0342$ |
| AWLPR | $0.6933 \pm 0.1120$ | $5.4492 \pm 3.5053$ | $0.8731 \pm 0.0581$ | $0.5031 \pm 0.1308$ | $21.7485 \pm 9.7197$ | $0.4326 \pm 0.0228$ |
| IW | $0.6235 \pm 0.1338$ | $5.9936 \pm 4.1849$ | $0.8851 \pm 0.0528$ | $0.5375 \pm 0.1205$ | $19.3256 \pm 8.5587$ | $0.4315 \pm 0.0355$ |
| GFTD | $0.6707 \pm 0.1356$ | $5.6733 \pm 3.8064$ | $0.7607 \pm 0.1582$ | $0.4773 \pm 0.1363$ | $21.2761 \pm 12.0001$ | $0.4188 \pm 0.0317$ |
| LPSR | $0.5696 \pm 0.1320$ | $6.6137 \pm 4.5335$ | $0.9070 \pm 0.0493$ | $0.6399 \pm 0.1134$ | $18.9086 \pm 8.1401$ | $0.4699 \pm 0.0315$ |
| Ideal | 1 | 0 | 1 | 1 | $+\infty$ | 1 |

[1,2] The values of indicator CC/SAM correspond to the results in the full spatial resolution.

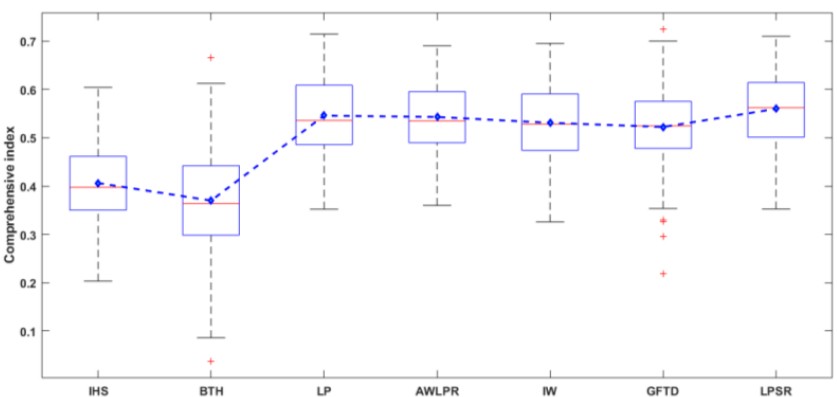

**Figure 8.** The box plot and mean values of the comprehensive index on a large set of images.

Based on the consideration of visual inspection and statistical analysis, the proposed LPSR fusion method can significantly enhance the spatial and texture characteristics of multispectral images, while preserving the spectral signature of ground features at the same time. Compared with the CS-based (IHS and BTH), MRA-based (LP-based and AWLPR), hybrid (IW), and GFTD methods, the LPSR method achieves the best balance between spectral characteristics preservation and spatial and texture features enhancement.

## 4. Discussion

### 4.1. Adjustment Capability

In practical usage, different application scenarios may have entirely different requirements on what the multispectral and SAR fusion images should be like. The flexible adjustability will make the fusion method more powerful and meet different needs in practical use. This section explores the adjustability of fusion results via tuning the decomposition level of the Laplacian pyramid to make the fusion results strike different balances between spectral preservation or spatial and texture enhancement. Therefore, the decomposition levels 1–5 are chosen for comparison experiment, and the corresponding fusion results are denoted as LPSR-1 to LPSR-5. In addition, the representative CS-based method (IHS fusion) and MRA-based method (LP-based fusion) are included as baselines of spatial enhancement and spectral preservation. Relevant experimental results are shown in Figures 9–11.

As can be seen from Figures 9–11, the fused image incorporates more characteristics from SAR image. Therefore, the spatial and texture enhancement is more significant when the decomposition level arises from 1 to 5. Nevertheless, the color changes more considerably when the decomposition level increases. When the decomposition level is 1 or 2, the fused image is visually similar to the LP-based fusion image in spectral consistency, while the spatial improvement is close to the IHS fusion method when the decomposition level reaches 4 or 5. Therefore, different trade-offs can be achieved between spectral fidelity and spatial improvement by adjusting the decomposition level of the Laplacian pyramid. From the statistical results in Figure 12, we can draw a similar conclusion. When the decomposition level equals to 1, the proposed LPSR fusion method achieves a good performance in spectral index CC, which is close to the LP-based method. While the decomposition level increases from 1 to 5, the spatial evaluation indexes (SCC and SSIM) of the proposed LPSR method approach the IHS fusion method, which indicates that the fusion result can be converted from the mode emphasizing spectral preservation to focusing on spatial and texture details injection. The verified adjustability makes the proposed fusion method more applicable and can satisfy different application requirements.

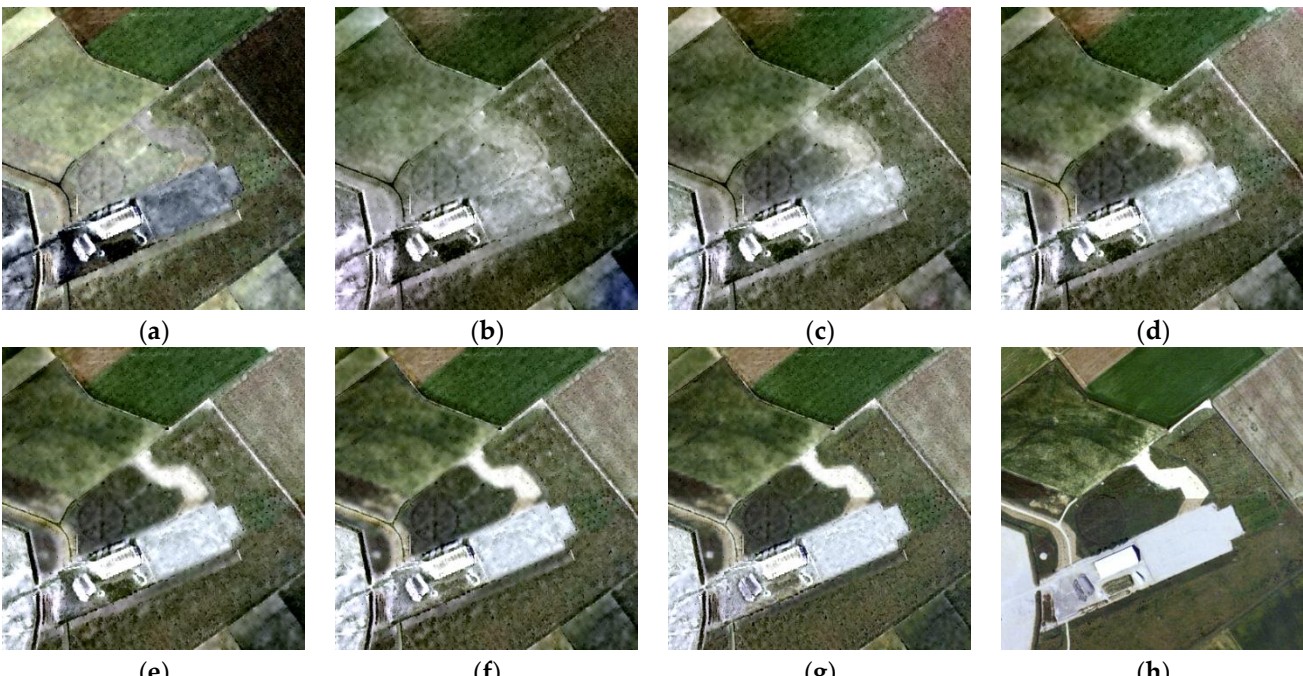

**Figure 9.** Fusion results of adjustability experiment in area 1. (**a**) IHS fusion; (**b**) LPSR-5; (**c**) LPSR-4; (**d**) LPSR-3; (**e**) LPSR-2; (**f**) LPSR-1; (**g**) LP-based fusion; (**h**) original HR multispectral image.

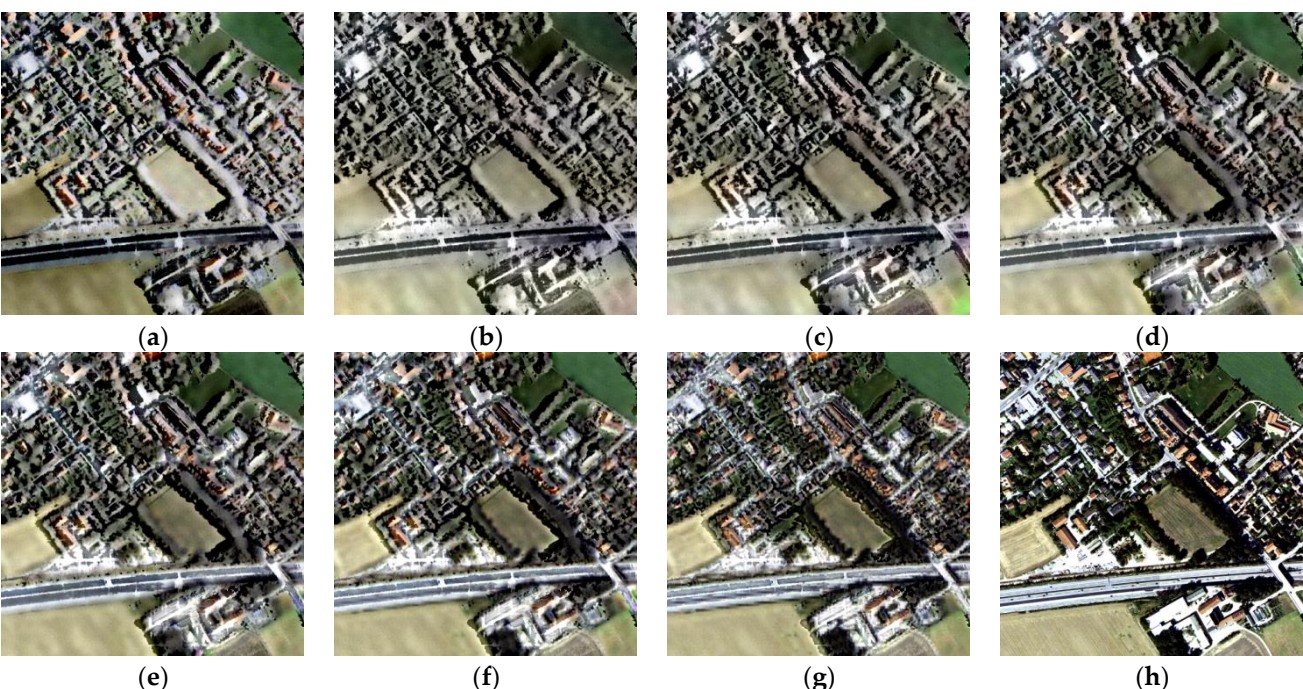

**Figure 10.** Fusion results of adjustability experiment in area 2. (**a**) IHS fusion; (**b**) LPSR-5; (**c**) LPSR-4; (**d**) LPSR-3; (**e**) LPSR-2; (**f**) LPSR-1; (**g**) LP-based fusion; (**h**) original HR multispectral image.

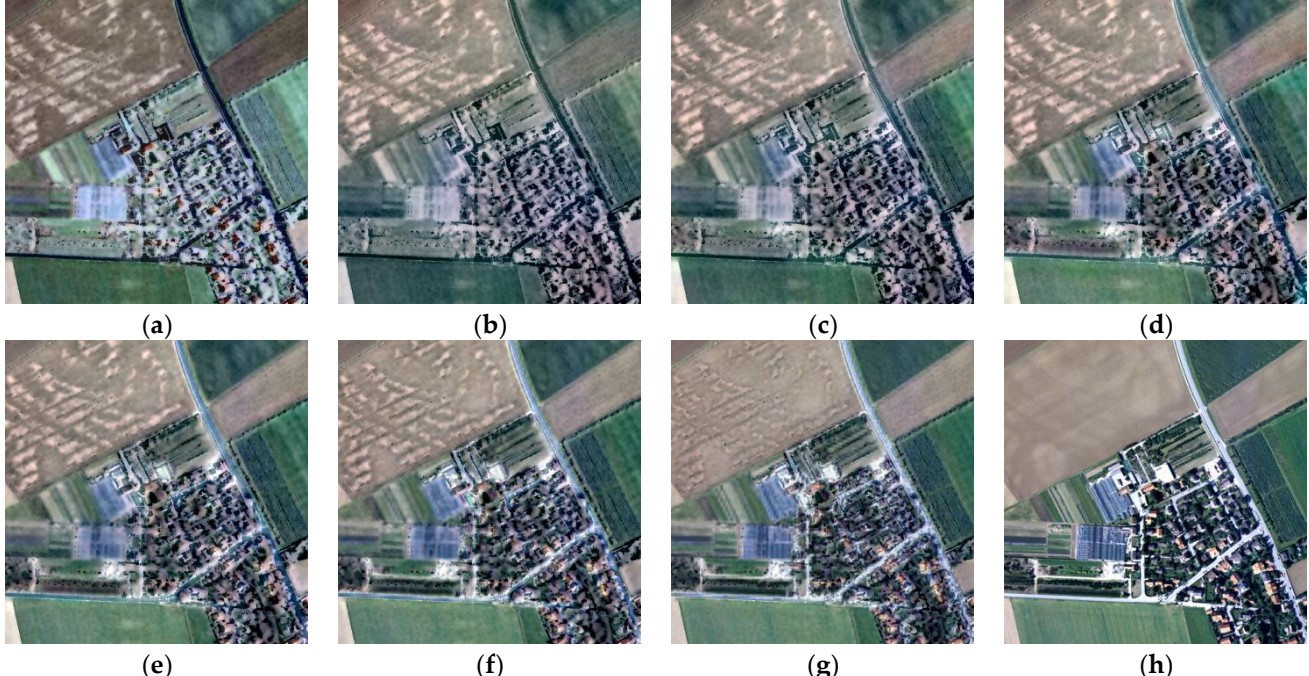

**Figure 11.** Fusion results of adjustability experiment in area 3. (**a**) IHS fusion; (**b**) LPSR-5; (**c**) LPSR-4; (**d**) LPSR-3; (**e**) LPSR-2; (**f**) LPSR-1; (**g**) LP-based fusion; (**h**) original HR multispectral image.

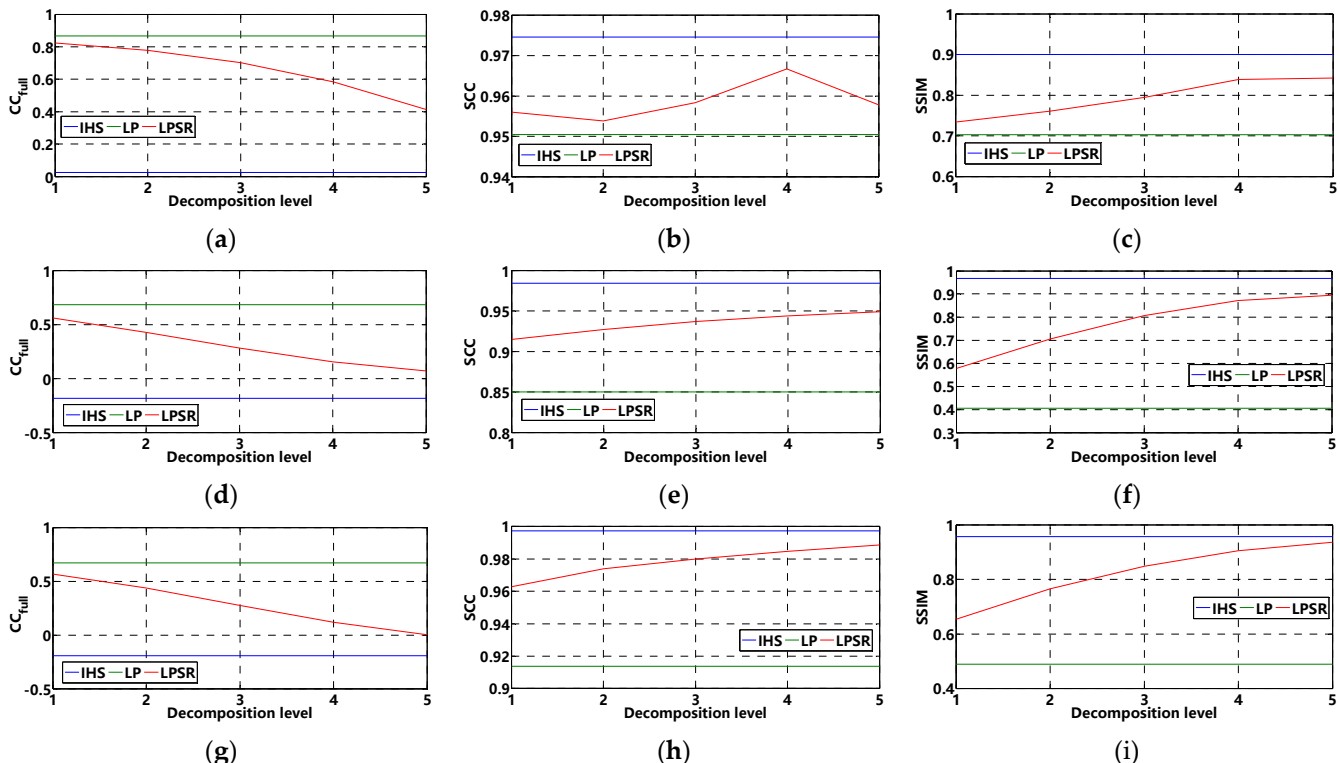

**Figure 12.** Statistical results of adjustability experiment: (**a**–**c**) display the $CC_{full}$, SCC, SSIM values of area 1; (**d**–**f**) for area 2; (**g**–**i**) for area 3.

### 4.2. Time Complexity

As mentioned before, the combined use of LP and sparse representation can overcome the drawback of the stand-alone SR-based fusion method because the most time-consuming stage, which is to say the sparse coding operation, is only performed in small-sized low-pass bands. In this part, we are going to discuss the relationship between time complexity and Laplacian pyramid level. All the fusion methods in this part are implemented in MATLAB 2020b on a computer with an Intel Core i9-10900K 3.7 GHz CPU and 128 GB RAM.

As can be seen in Table 5, the stand-alone SR-based method consumes a high computing burden. Our proposed method with only one decomposition level can also sharply reduce the running time to some extent. As the decomposition level increases, the low-pass component has a smaller size, leading to a lighter sparse coding operation. Therefore, the time complexity is also reduced. The running time of proposed method with decomposition level 5 is comparable to those of the traditional methods. This part confirms that the computational efficiency can be improved when sparse representation is combined with LP.

**Table 5.** Running time (in seconds) of the proposed method with different LP levels.

|        | SR      | LPSR-1  | LPSR-2 | LPSR-3 | LPSR-4 | LPSR-5 | GFTD   | IW     | AWLPR  | LP     | BTH    | IHS    |
|--------|---------|---------|--------|--------|--------|--------|--------|--------|--------|--------|--------|--------|
| Time/s | 9308.66 | 2226.38 | 536.63 | 119.59 | 23.36  | 2.95   | 0.2896 | 0.1137 | 0.2047 | 0.0830 | 0.1362 | 0.0234 |

### 5. Conclusions

This paper proposes a fusion method combining the Laplacian pyramid and sparse representation. Experimental results demonstrate that the proposed method strikes the balance between spectral preservation and spatial enhancement in terms of visual inspection and statistical comparison. The fusion framework integrates the merits of multi-scale transform and sparse representation theory, leading to the prevention of artifacts and inclination to smooth details. Additionally, the capability of achieving different balances between

spectral consistency and detail improvement extends the applicability and practicability of the proposed method. Evaluation of pixel-level multispectral and SAR image fusion is challenging, considering that data fusion is more or less data-driven and application-oriented. Therefore, future work will concern more about mapping and classification tasks with and without fusion strategy, in order to validate the advantages of products derived from multispectral and SAR image fusion over datasets generated by stand-alone data sources.

**Author Contributions:** Conceptualization, H.S. and H.Z.; methodology, H.Z.; software, H.Z.; validation, H.Z.; formal analysis, H.Z.; investigation, H.Z.; resources, H.Z. and H.S.; data curation, H.Z.; writing—original draft preparation, H.Z.; writing—review and editing, H.Z., Q.Y., H.S., and X.G.; visualization, H.Z.; supervision, H.S. and Q.Y.; project administration, H.S.; funding acquisition, H.S. All authors have read and agreed to the published version of the manuscript.

**Funding:** This research was supported by National Natural Science Foundation of China (NSFC), grant number 41971303.

**Institutional Review Board Statement:** Not applicable.

**Informed Consent Statement:** Not applicable.

**Data Availability Statement:** The data presented in this study are available on request from the corresponding author.

**Acknowledgments:** We would like to thank Olaf Hellwich (Berlin University of Technology) for providing the datasets for experiments. We are also grateful to the editor and the anonymous reviewers for their detailed review, valuable comments, and constructive suggestions.

**Conflicts of Interest:** The authors declare no conflict of interest.

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
