# Peer review of "Multispectral and SAR Image Fusion Based on Laplacian Pyramid and Sparse Representation"

_remotesensing, doi:10.3390/rs14040870_

Round 1
Reviewer 1 Report
This paper proposes a novel fusion method, comments are as below.
- Please highlight the motivation of this fusion work. You might give some qualitative results by comparing the fusioned images with SAR/multispectral images to highlight the differences and the advantage of doing such fusion.
- I cannot find any information from Fig. 2, actually this figure contains no useful information. I want to see a meaningful flowchart.
- Please compare the methods with some deep learning-based methods.
- If possible, you may also refer to some transfer learning way and cite them: a. DT-LET: Deep transfer learning by exploring where to transfer, 2020. b. Dual adversarial network for unsupervised ground/satellite-to-aerial scene adaptation, 2021. c. Unifying Top–Down Views by Task-Specific Domain Adaptation, 2020.
Reviewer 2 Report
The Authors undertook to describe a very difficult and important issue, which is undoubtedly the fusion of multi-spectral images and SAR images. Perhaps even more interesting, they used Laplacian Pyramid and Sparse Representation to improve the proposed method of image fusion.
As a result, the reader will be pleased to analyze how complementary information from multi-sensors can be combined to improve the availability and reliability of stand-alone data, leading to more efficient edge or cloud computing decisions. The authors consider multi-spectral images which contain plentiful spectral information of the Earth's surface that is beneficial for identifying land cover types. Besides, synthetic aperture radar images can provide abundant information on the texture and structure of target objects. These advantages of individual types of images can be combined and as a result a synergy effect can be expected by integrating the information from MS and SAR images. How Laplacian pyramid can decompose both the multi-spectral and SAR images into high-frequency components and low-frequency components is skillfully shown. The result obtained the effect that different processing strategies can be applied to multi-scale information. On the other hand, low-frequency components are merged based on sparse representation. In contrast, high-frequency components of images are combined based on a certain activity-level measurement identifying salient features. Moreover, Laplacian pyramid reconstruction is successfully performed to obtain the integrated image. Experiments on several datasets confirmed the effectiveness of the proposed method. The authors have correctly demonstrated that both visual interpretation and statistical analyzes demonstrated that the proposed method strikes a satisfactory balance between spectral information preservation and enhancement of spatial and textual characteristics. It is also worth emphasizing that the proposed approach has a high level of flexibility for several application scenarios.
However, there are minor shortcomings in the work that should be corrected.
1. The introduction should clearly define the scientific purpose of the manuscript and the contributions - it is best to separate one consistent paragraph per this passage.
2. All the markings in Figure 1 should be explained in the text - I did not notice the description of N.
3. What is the comlexity of the proposed method?
3. Some typos:
a) line 422 -> next page
b) a reference to Table 2 is below it, and should be before it is embedded in the text.
The above comments do not detract from the high grade of the manuscript, and after the Authors have corrected them, the manuscript may be published.
Reviewer 3 Report
This manuscript presented a multispectral and SAR image fusion method based on Laplacian pyramid and sparse representation. The method combined the merits of Laplacian-pyramid-based method and sparse-representation-based method. The results are valuable and are properly discussed. The structure is well-organized and the authors demonstrated a good knowledge of the problem and of the related literature. Overall, I think this work can be published with minor revisions. My comments on this manuscript are as follows:
1. The main contributions can be highlighted in the Introduction section to make it clearer for readers.
2. Line 252, the comprehensive index (CI) should be clarified in Section 3.2. The metrics and weights used and the reasons.
3. Page 16, the quality of Figure 11 should be improved to make the green line in Figure 11(b) more readable.
4. The references should be revised according to the styles of MDPI, especially for the conference proceedings, like Ref. [18] [44].
5. Grammar needs slight changes, like image or images (Page 11, Line 342), with or without the (Page 3, Line 105).
Reviewer 4 Report
In the paper, fusion approach in which multispectral and SAR images are combined using sparse representation and Laplacian pyramid:
1. The contribution of this study seems superficial as the Laplacian Pyramids and sparse representations are quite old techniques, used together many times in many subfields of computer vision. Hence, the main findings of this study are quite expected. Where is the effort? This should be clarified. What was overlooked in previous studies?
2. Are the employed three pairs of images (line 222, page 6) enough to conclude on the superiority of a given technique? The study should involve quantitative tests.
3. The methods that were used in experimental comparison are outdated: [11] (1990) [13] (1992)[17] (2002)[20] (1985) [27] (2009) [51] (2016). Note that the method described 20 years in [20] also uses the Laplacian pyramid.
4. The evaluation metric should include more domain-specific algorithms as the SSIM (page 7, line 245) is a representative of quality assessment methods designed for natural images. Here, the erreur relative globale adimensionnelle de synthese (ERGAS), the universal image quality index, and the universal image quality index of pan-sharpened multispectral imagery (q4) would be a better choice.
5. The running time and high-computation demands are mentioned (Sections1-2) but there are not addressed in the experimental section. The identified weaknesses should be demonstrated along with the proposed improvements.
6. The results cannot be replicated by a reader.
Reviewer 5 Report
This paper proposed a fusion framework to integrate the information from MS and SAR images based on Laplacian pyramid (LP) and sparse representation (SR) theory. LP is performed to decompose both the multispectral and SAR images into high-frequency components and low-frequency components, so that different processing strategies can be applied to multiscale information. This is an interesting research paper. There are some suggestions for revision.
- The innovations of this paper are not clear. Why do you want to do this? What are the advantages of the proposed solution? It is recommended to focus on the related discussion. Please highlight the innovations/contributions of the proposed solution in introduction.
- Authors ignore some relevant papers, such as "A novel multi-modality image fusion method based on image decomposition and sparse representation", Information Sciences 432, 516-529, 2018, "An image fusion method based on sparse representation and sum modified-Laplacian in NSCT domain", Entropy 20 (7), 522, 2018, and "A novel fast single image dehazing algorithm based on artificial multiexposure image fusion", IEEE Transactions on Instrumentation and Measurement 70, 1-23, 2021. Please discuss them carefully.
- Please specify which issues are solved by the proposed solution and which aspects of existing solutions are surpassed by the proposed solution.
- Section II lacks necessary descriptions related to graph, theory, and formula analysis. The readability of formulas and theoretical descriptions is poor.
- The experimental results are not convincing. Please compare the proposed solution with more recently published solutions.
- What is the experimental environment?
- Please specify how to obtain the suitable parameter values used in the proposed solution.
- The running time is mentioned in introduction. Please discuss the time complexity of the proposed solution.
Round 2
Reviewer 4 Report
In the paper, a method for multispectral and SAR image fusion is presented. The revision improved the quality of the paper and clarified some issues. Comments:
1. The complexity analysis should also cover the run time reported for other approaches to show their efficiency.
2. The paper should contain a link to a webpage where the future source code will be located (mandatory). The readers should be able to replicate the results. Also, this would largely promote the paper.
3. I agree that many papers do not contain quantitative analysis; however, such drawbacks should not be used as an excuse since a method may favor some sets of images. If experiments on a larger set of images, and reporting average values with a standard deviation of quality indices, are not possible, please provide a discussion on the stability of the results disregarding the variability of input images.
Reviewer 5 Report
All my concerns have been addressed. I recommend this paper for publication.
Author Response
Thanks for your comment. We appreciate all your suggestions and comments, which really help improve the quality of the manuscript.